# Pipeline Neighbors: How Can We Avoid Conflicts?

**Natalya Novikova**

The Institute of Ethnology and Anthropology, Russian Academy of Sciences, Leninsky Pr., 32A,
119991 Moscow, Russia; natinovikova@gmail.com; Tel.: +7-9161903860

**Abstract:** In this article, I consider the various policies of oil and gas companies relating to Indigenous peoples of the Russian Federation. The analysis is based on field research in Northern Russian regions. Data for the analysis comprises: International standards, Russian laws, corporate codes, official regulations, and interviews with company employees and representatives of the Indigenous populations. The research methodology is based on the concept of legal pluralism as the coexistence of various legal regimes and the search for platforms of common interests. The goal of this article is to analyze policies on benefit sharing by assessing projects and programs adopted by various industrial companies according to the social and humanitarian prospects of their social acceptance. I consider the possibilities for Russian legislation to promote respecting Indigenous people's interests in the preparation of corporate sustainability reports.

**Keywords:** indigenous people; oil and gas companies; corporate social responsibility; Russian North; benefit sharing; legal regulation

## 1. Introduction

Achieving partnership between Indigenous peoples and industrial companies in the Russian North is important not only for the survival of such populations, but for operation of major economic sectors, and the social and economic development of the Arctic and regions of the Russian North. Today, awareness regarding mutual responsibility and mutual interest in the establishment of such new relations between companies and Indigenous communities is only beginning to arise. Of special importance to their relations with industrial companies are the consultations with Indigenous peoples, and their ability to say "no" to projects that violate their rights and legal interests. New international documents have been adopted that govern the processes I consider in this article; indeed in Russia the awareness of the need to address this problem is growing. Economic and sociological research focus much attention on the issues of compensation, and thus the distribution of company profits to society. In this article I focus on the non-material/non-financial aspects of mineral extractivism in areas inhabited by the Indigenous peoples of the Russian North, and on possible means of their regulation.

### 1.1. Main Themes, Objectives, and Methodology

The goal of this article is to analyze policies on benefit sharing. The major research questions are: How has the social climate in regions of industrial development changed over time? How has the experience of living next to each other—of living as neighbors—come to shape attitudes of both representatives of industrial companies and members of Indigenous communities towards monetary compensation?

The study is based on legal anthropology methods, which combine ethnographic field research with the analysis of national laws and internal corporate documents. The benchmark for comparison comprises international standards [1,2] and Russian standards for business and sustainable development [3,4]. As a framework I employ the concept of legal pluralism, which acknowledges the

coexistence of various legal regimes [5,6]. It is this very coexistence that ensures a relatively peaceful cohabitation, while domination or coercion by one party provokes conflicts. These approaches allow a more comprehensive consideration of all the actors involved in environmental management and highlight the opportunities for diverse regulations of the relations between Indigenous peoples and industrial companies.

The methodological approach of comparing the ideal and experienced legal order is useful for my work [7]. Since I focus on normative culture, my method incorporates an anthropological critique of legislation and corporate documentation. I have first collected and studied the documents and regulations, which lay out the rules of behavior for workers, and then, during field research, considered the extent to which these are lived out with respect to Indigenous people; they are experienced by the latter in everyday life.

So far, no country has created an ideal model of benefit sharing with regard to oil and gas production and impacted Indigenous communities. Research of the interrelationships between Arctic communities and industrial companies demonstrates, "Indigenous people seek jobs and income while not undermining opportunities for using local resources and have enduring nutritional, cultural and economic benefits." This tension reflects the definition of sustainable development and also self-determination and self-reliance of Indigenous people [8] (p. 122).

The academic and public discourse is increasingly acknowledging the need to study all environmental stakeholders and manage cultural diversity, especially in large federal states. A comparative study of benefit sharing issues associated with industrial development stresses, "the importance of communication, cooperation and consultation [...] This is crucial not only for Indigenous peoples—who need to be active partners from the beginning of access and benefit sharing initiatives, and to receive clear and honest information about projected benefits and risks—but also for governments, which have a responsibility to establish simple and clear information channels, to cooperate among themselves where resources or knowledge crosses national borders, and to establish transparent and fair consultation processes for the development of access and benefit sharing policies" [9] (p. 349). The former UN Special Rapporteur on the Rights of Indigenous Peoples, James Anaya, has discussed the interactions between Indigenous peoples and industrial companies. One of his reports is dedicated to the status of Indigenous peoples in the Russian Federation in the context of industrial development. Anaya noted: "In many instances agreements have been concluded for the development of natural resources on or near Indigenous lands, bringing some benefits to the Indigenous peoples concerned. However, a broader understanding of cooperation is called for: Rather than limiting the interaction between extractive industries and Indigenous people to compensation agreements, administrators should encourage ownership interest and profit-sharing in extractive industries, when Indigenous communities are so inclined. The federal government should establish reliable methods of monitoring the development of industrial projects…" [10]. So far, these recommendations have been mere suggestions for Russia.

The Indigenous peoples' level of involvement in resource co-management, decision making, obtaining compensations, and profit sharing is largely dependent on constitutional guarantees and land tenure security. In northern Canada, the interaction process is enshrined in the Constitution Act, 1982 and in the agreements with the federal government. Also, Indigenous peoples have had to advance their rights and interests through the Canadian courts. As a result, for example, a long-standing policy development process took place in the Northwest Territories of Canada regarding the construction of a pipeline in the Mackenzie valley. This led to the fact that "Indigenous peoples are major stakeholders in Mackenzie Gas Project. Aboriginal Pipeline Group expect to receive additional capacity, which will eventually increase its ownership percentage to 33.33%" [11] (pp. 108–109). One Indigenous leader said: "A pipeline down the Mackenzie Valley will…not destroy the land, but without some form of economic base we will surely destroy our people" [11] (p. 111).

The example of Alaska demonstrates the joint-stock model of profit sharing and the difficulties it has created for the social and economic development of local communities [12]. Researchers note: "As the scale of industrial development increases, however, costs to landscape, wildlife, subsistence

and cultural values can also increase while benefits from jobs and immediate income remain modest or even decline" [8] (p. 134).

Indigenous peoples give preference to working with local sustainable resources, usually seeing industrial development as a risky business with high expectations that cannot always be fulfilled. Therefore, benefit sharing may only mitigate the negative consequences of industrial development. In Russia, these practices look even more precarious, especially since industrial companies and government authorities practice benefit sharing primarily in the form of financial compensation.

In the past few years quite a number of papers on corporate policy have been published, including one by this author [4,13–15]. I approach the problem by looking for effective corporate and social policies in the Russian North as well as opportunities for legal regulation of benefit sharing. I will reflect on the fact that many oil and gas workers, who were formerly fly-in/fly-out employees, became local residents, and I will examine how this fact impacts benefit sharing. I will also consider who implements the policies of industrial companies and how.

## 1.2. Legal Regulation

In Russia, relations between Indigenous peoples and industrial companies are primarily regulated by constitutional and legal provisions in the special laws on rights of Indigenous peoples [16]. The Federal Law 'On the Guarantees of Rights of Numerically Small Indigenous Peoples in the Russian Federation' defines the right to compensation (collective and individual) as reimbursement for damages [17]. Unfortunately, these provisions are not incorporated in the legislation on resources, thus limiting their capacity to a large extent. Article 42 on profit distribution, which was incorporated in the first version of the Federal Law (1995) 'Subsoil Law', was abolished in a later stage. Federal Law 'Subsoil Law' (1995 edition) Article 42, "During the development of underground resources in the areas inhabited by small Indigenous peoples and ethnic groups payments to the budgets of the subjects of the Russian Federation shall be partly used for the social and economic development of these peoples and groups." [18]. Indigenous organizations and experts have for many years tried unsuccessfully to promote the protection of the Indigenous people's rights and to introduce corresponding amendments to this law. The Federal Law 'On Production-Sharing Agreements' guarantees reimbursement for damages to Indigenous peoples [19]. Article 7 of this law stipulates the responsibilities of investors operating within areas of traditional residence and economic activity of numerically small Indigenous populations to take measures to protect the latter's ancestral lands and traditional way of life and to appropriately compensate them. In 1994, Sakhalin Energy signed such an agreement to implement the Sakhalin-2 project. The same principle applied to Sakhalin-1 project (1995), implemented by Exxon Neftegas.

To a large extent the process of lawmaking at a Federal level was based on the legislative potential of the regions. So, for many of these regions this process of lawmaking was even more constructive then for those at the center. Nevertheless, legislation is not itself enough to protect Indigenous peoples [20].

In the Russian Federation, there are also 'Methods for Calculating the Scale of Damages', approved by a 2009 decree of a no-longer existing ministry, which are, nevertheless, still in place. This document has been criticized often for a number of reasons including the following two: (1) The methods only apply to legal entities, (2) the amount thus calculated may be less than the compensations the companies already paid. For instance, such criticism was made at a conference in Khanty-Mansiysk in 2018 [21]. Today the Federal Agency on Nationality Affairs and the Committee on Issues of Nationalities of the State Duma are working out new methods for calculating losses.

Insufficient legal regulation of issues concerning relations between Indigenous peoples and industrial companies, especially in relation to compensations and reimbursements for damages, as well as to minimization of negative consequences, leads to the necessity of considering corporate social responsibility standards [1,4].

## 2. Materials and Methods

The article is based on published research, academic literature, analysis of Russian legislation, and the empirical material that I collected in various regions of the Russian Arctic and Russian North. The fieldwork included participant observation and interviews with Indigenous residents, representatives of oil and gas extraction companies, and government authorities. The interviews were conducted in line with a specially developed guide aimed at revealing each interviewee's attitude to benefit sharing of oil and gas production. My key Indigenous informants were men and women aged 40–60 as well as youth living in close proximity to industrial development areas.

The article also draws on my experience as an expert on Indigenous peoples in the legislative assembly of the Russian Federation (2011–present). Every interview used in the article was conducted with free, previously informed consent.

One of the field work periods underpinning the article was conducted yearly between 1995 and 2000 in the Khanty-Mansi Autonomous Okrug, specifically in the city of Khanty-Mansiisk and in the villages of Surgutsky and Nizhnevartovsky regions, in Khanty and the Nenets camps, as well as among engineers and workers of the oil companies Lukoil-West Siberia and Surgutneftegas. In 2014, field work was carried out in both companies' headquarters in the cities of Kogalym and Surgut. Another field study was conducted in 2008–2012 in Yamalo-Nenets Autonomous Okrug, specifically in Yamalsky and Tazovsky regions and in the city of Salekhard, among various Nenets groups, as well as Gazprom and Lukoil employees. The next field work period was carried out in 2006, 2011, 2014, and 2019 in Sakhalin Oblast, namely in the city of Yuzhno-Sakhalinsk as well as in Nogliksky and Okhinsky regions. The study involved Indigenous inhabitants and local residents as well as Sakhalin Energy and Exxon Neftegas employees.

The article also makes use of the empirical material I collected in 2006 in the Northwest Territories of Canada during the public hearing on the Mackenzie Gas Project in the village of Inuvik (Scientific Research Licence # 14079N, Fale #12410684 Aurora Research Institute—Eurora College). In addition, the study draws on field data collected in 2018 while organizing a seminar in Bulun region (ulus) of the Sakha Republic (Yakutia).

The research topic requires establishing a relationship of trust between the researcher and the informants, which was ensured by my long-standing contacts in the field and the numerous phone consultations with my informants that followed the field stage of the study.

## 3. Results

### 3.1. How Oil Worker 'Migrants' became 'Neighbors' and What This Means for Indigenous Peoples

My research on this topic started in Khanty-Mansi Autonomous Okrug (KhMAO) in the 1990s; later I continued it in Yamal-Nenets Autonomous Okrug (YaNAO) and Sakhalin Oblast, which allowed me to observe the dynamics of company–community relations. At the early stage of industrial development, according to the interviews, Indigenous peoples were confident that presence of the arrived geologists, and later oil workers, is temporary, that they would work for a while and then return home, and that afterwards local life will return to its former state. At the same time, locals perceived newcomers as carriers of new information. They (especially, young people) liked how the latter sang guitar-strumming songs, handled cars, and shared stories. This process of acquaintance proved to be mutual. Back in 1920s some researchers wrote that, "without the reindeer and dogs, without an experienced native guide, our investigation in the Russian North and discoveries of new mining wealth would have ceased" [22]. In practice, in the absence of legal regulation and corporate social policy, the companies and Indigenous people were wary of one another, and relations were revolving mostly around a 'local–migrant' dilemma. During interviews recorded in KhMAO, the Khanty and Nenets often said that oil workers will leave, but the harm they have caused to the environment, to nature, in fact to the 'home' of the Indigenous people, will remain.

On the other hand, oil workers emphasized that this land has come to be a home for them: Their children live and study there and they are staying permanently in the newly built cities. The line was drawn between 'town people' and 'forest people'. In the 1990s, the Khanty, Mansi and Nenets started returning to their camps, securing 'clan lands', as they considered them their own. Those, who have

arrived to work for the oil industry by that time have already been settling in cities, such as Surgut, Kogalym, etc. They have also begun to consider the land they worked on as their own. An employee of LUKOIL-West Siberia told me: "We work here, we brought in the equipment, and then out of a sudden, from who knows where, these Nenets appear with their reindeer." From Yamal nomads, I frequently heard that oil-workers fly over the tundra when reindeer herders are pasturing their reindeer elsewhere and, seeing no structures on the ground, they assume that nobody lives there. Here I draw attention to this perception of land, historically used by Indigenous people, as terra nullius, because it initially hampered the process of dialogue and the understanding of a need to pay compensations.

Initially, oil companies showed little interest in the local population, and even less in the Indigenous peoples. They perceived themselves as champions of civilization, a blessing for the area, and thought very highly about their own social status. In recent years, changes in the mutual perception of both groups are taking place. Life in a common area next to the local population is becoming an important trait of the oil and gas community. The previously present mutual 'exoticization' is giving way to the perception of both oil-workers and the Indigenous peoples as being neighbors.

In recent years, many of the companies under consideration have developed special policies related to Indigenous peoples, which are reflected both by corporate websites and sustainability reports. The Russian Union of Industrialists and Entrepreneurs (RUIE) publishes corporate practice guidebooks. In 2017, a survey was carried out among RUEI members on sustainable development goals (SDGs). The main activity area of companies, which helps achieve the SDGs, according to two-thirds of those surveyed, was the introduction of innovations and new energy-and resource-saving technologies. The second most popular answer—'stable economic growth—63.3%, and growth in employment' was chosen by 56.7% of the respondents. Just a little less—53.3% chose the answer 'responsible use and production'. Supporting a dignified way of life for workers and members of their family was noted by 46.7% of those participating in the survey. About the same number chose the answer "contribution to sustainable development of the territory under development, the improvement of the social climate, the acceleration of economic growth." Decreasing negative ecological influences, and climate impacts as was noted by 33.3% [4] (p. 10). Sakhalin Energy, Gasprom Neft, LUKOIL and some of the other largest fuel and energy companies have been traditionally recognized as leaders here. Many of these companies are also leaders in terms of their policies regarding relations with Indigenous peoples in their areas of operation, as confirmed by annual awards. For instance, since 2000, LUKOIL–West Siberia has received diplomas for collaboration with Indigenous peoples.

At the same time, both at the early stage of industrial development and nowadays, Indigenous persons pursuing traditional activities experience stress from being neighbors with oil workers. Some Khanty and Mansi people actively protest; others remain apathetic, having no faith in possible changes. Linked to this is the suspicion of and disregard for ethnological assessments on the part of the Indigenous population. Even if they assist the researchers, they often think that companies will continue doing their business nevertheless, once having arrived. Field research shows that in the absence of an effective policy, industrial companies will never become 'good neighbors' to Indigenous peoples.

*3.2. Case Studies of Benefit Sharing Regimes: The Diversity of Oil and Gas Companies' Policies Regarding Indigenous Peoples in Russia*

Usually industrial companies' policies regarding Indigenous peoples are evaluated by the existence or absence of relevant agreements and compensation payments. It has become somewhat of an academic tradition in publications to compare corporate policies in the KhMAO, YaNAO and Sakhalin Oblast, so I will limit myself to a brief overview.

Khanty Mansi Autonomous Okrug-Yugra (KhMAO)

KhMAO's distinctive features are a well-developed legal system pertaining to the numerically small Indigenous peoples and their relations with industrial companies; the allocation of about 500 regional-level Territories of Traditional Nature Use (TTPs) to these peoples (families and clan communities (Clan community is an economic unit of individuals representing, mostly, Indigenous numerically small peoples of the North, Siberia, and Far East of Russia. In practice, since their establishment clan community have been playing the role of economic enterprises employing not only relatives and kinsmen, but also friends and acquaintances).

Since the 1990s, economic agreements have existed between industrial companies and Indigenous peoples (originally the heads of clan lands, later the subjects of rights to traditional use of natural resources). The regional law "On Territories of Traditional Nature Use of the Numerically Small Indigenous Peoples of the North in Khanty Mansi Autonomous Okrug—Yugra" stipulates that industrial activity on such territories requires approval of natives to develop production sites. Should they disapprove, a special commission shall examine the production site documents, which will share its recommendations with the Okrug government. Basically, from the legal perspective the general pattern seems fair, but practice shows that the commission's decisions favor oil workers. Another model was developed in the 1990s, namely a special environmental management regime at certain extraction fields, in particular in the Surgut area. In practice, the territories inhabited by Indigenous people became reservations, with limited access for the population [10]; this practice of restricting the movement of Indigenous residents continues today, according to social media reports. In 2009, the Okrug adopted a Model Agreement by government decree, which defines the scope of agreements signed with Indigenous peoples. Every family and community dealing with oil workers are listed into the Register of Subjects of Traditional Nature Use, which is a guide for oil workers.

This current system features a number of obvious disadvantages: The relationships are individualized, and in the event of conflict, they take on a private nature of a bilateral agreement with a large industrial company. According to this scheme, government authorities are excluded from such relations. Besides, there are no powerful non-governmental organizations; today, they often agree to the industrial companies' policies. The current practice satisfies the parties until companies claim new lands, especially in the Surgut and Nizhnevartovsk regions. Then, many conflicts arise in relation to reindeer pastures or sacred sites.

The largest industrial companies operating in the area have adopted corporate documents specifying their policy on Indigenous peoples; they also have departments on Indigenous people's relations. Responsible representatives of companies sign agreements with Indigenous people, track their implementation, and provide funding for various activities and programs. Over decades of industrial development, the policy of oil companies has changed significantly, but even now it is carried out residually, and oil companies sometimes freely shift the terms of payment or services for the next year.

Consider the policies of two largest companies operating in the Okrug. In spite of the existing legislation and documents adopted by the regional government, for instance, the Model Agreement, companies behave differently. LUKOIL mainly adheres to international standards; this company joined the Global Agreement Network in Russia and often receives awards in competitions for working with Indigenous people's relations and for sustainable development, etc. The company's policy is aimed at maximizing conflict prevention and its staff often provide informal support to Indigenous people, providing them with fuel, food, transportation, and technical assistance.

Surgutneftegas is the only major oil company that has its headquarters in the Okrug. Both local authorities and people recognize its authority and role in the development of local territories. However, in these days the company's relations with Indigenous communities have been characterized by conflict. Surgutneftegas insists on a rigid adherence with legislation (as it interprets it) as the only blueprint for action. In particular, company lawyers insist they are entitled to decide who represents Indigenous peoples and thus can qualify for the signing of economic agreements. Such harsh policy provokes dissatisfaction among the Indigenous peoples. This company is involved in many conflicts over the preservation of sacred sites of Indigenous peoples. It can be said that this

is an outcome of Surgutneftegaz's disregard for international norms and standards, and generally for ethical standards of behavior.

A significant problem in relations between Indigenous peoples and industrial companies is the difference in their perceptions of justice. Among permanent residents of the area, brought together and employed by Surgutneftegas and LUKOIL–West Siberia, (a division of LUKOIL)—the engineers and workers—widespread is the opinion that payments to natives is a social injustice. They believe that companies pay enough taxes, and it is the job of the government to provide social support for Indigenous peoples. Exclusion of public authorities from the process of interaction between Indigenous peoples and industrial companies under a Model Agreement intensifies such conflicts. In short, the main disadvantage of the policies of companies in the KhMAO is their focus on conflict resolution and financial support, and not on the sustainable development of the Indigenous peoples of the North.

Yamalo-Nenets Autonomous Okrug (YaNAO)

The YaNAO occupies a special place among the regions of the North, since the nomadic population living there preserves a traditional way of life based on reindeer herding. The significant percentage of nomads (primarily in the Yamal and Taz districts) contributes to the adoption of important legal documents in the district. Plans to sedentariness and to resettle into villages the reindeer herders proved to be meaningless, as anthropologists had predicted, but there are constant proposals, in one form or another to reduce herds. The issue of how to combine traditional nature use and industrial development in the YaNAO has recently become particularly relevant: How it will be solved will affect both the lives of Indigenous peoples and the geopolitical interests of Russia, which is concerned not only in industrial development, but also in having a population in its arctic territories.

The general demand of Indigenous organizations in YaNAO is the transition from corporate charity to the establishment of partnerships and cooperation in matters of mutual interest. Industrial enterprises to some extent also seek ways to diminish the impact of industrial development on the Indigenous peoples, to avoid open confrontation. They sign agreements with the Okrug Government and local (rayon) administrations. Based on these agreements, companies transfer funds, but there is no effective control over their distribution and use. Disadvantages of this system are the lack of targeted funding, transparency, and public (Indigenous) participation. Field research shows that against the background of general environmental problems related to industrial development and climate change, companies often allow themselves non-compliance regarding current legislation and their own obligations. The current practice of setting penalties that are insignificant for the fuel and energy companies also contributes to such an approach.

When considering interaction with industrial companies, we must take into account the fact that industrial development gives different groups of the population unequal opportunities and leads to different consequences. Tundra people experience a reduction of pastures and environmental pollution. Villagers, both Nenets and other residents, await new jobs from the development of industry in the north. If companies develop infrastructure in the towns, the local population, including the Nenets, would support industrial projects enthusiastically.

To enjoy psychological well-being, nomadic reindeer herders need to be confident that they can continue their activities in the future. Many Nenets have already made their choice—reindeer herding is their occupation, work, and way of being. Today, when the process of approving industrial development is underway, Indigenous people often do not believe that their opinion will be taken into account, especially since the organizations of Indigenous peoples often receive documents for approval only after all others have signed them. This exposes the Nenets to psychological pressure.

Another negative aspect is that not every reindeer herder and fisherman knows about the construction details in advance, so they do not know what they are going to receive or lose. Insufficient information provokes rumors and conjectures that all Nenets people will be relocated to villages, that they will not be allowed to nomadize with their reindeer, etc. People resent that the industrial development is permitted in the elevated tracts of land where the Nenets people locate their camps and have holy sites. Experts and municipal authorities believe the development of the

natural resources of Yamal should follow a systematic approach, built on the basis of interaction of all stakeholders—local state institutions, industrial enterprises, and Indigenous peoples. Today, contractual and financial practice approach prevail, without taking into account alternative development paths. This has a negative impact on the situation of Indigenous peoples and has reduced the legal regulation of companies in the region. Legislation monitoring shows that in recent years, there has been a significant change in the legislation of YaNAO and most of the previously adopted documents have ceased to operate. Many regional documents merely duplicate federal ones, featuring no provisions on Indigenous people's rights. The exclusion of such provisions from new laws on local referenda and cultural heritage sites negatively impacts the Indigenous people's status. Unfortunately, instead of adopting a law on social impact assessment, as planned in 2008, a weak norm appeared in the law 'On the protection of the ancestral lands and traditional way of life of Indigenous Numerically Small Peoples of the North in the YaNAO' that stipulates "scientific and sociological research for impact assessment purposes." Awkward wording is not so important in this case, as to recognize that the Okrug legislation takes a step back in terms of protecting Indigenous people's rights. Current YaNAO legislation seems rather like governmental support measures targeting peoples who lead a traditional way of living, mostly in material and financial spheres. Thus, government bodies are moving further and further away from monitoring the activities of industrial companies and regulating their relations with Indigenous peoples. Under these conditions, restrictions on economic activity and compensation due to Indigenous peoples that are guaranteed by law hang in the air, without law enforcement mechanisms and with contributing to the sustainable development of the region.

Sakhalin Oblast

In Sakhalin Oblast, several variants of interactions between energy companies and Indigenous and local populations emerged historically, and the previous experience of corporations and their successors reflects in the present. The oldest among the existing companies is Sakhalinmorneftegas; Rosneft and Gasprom have started operating in this region relatively recently. International companies Sakhalin Energy and Exxon Neftegas Limited also operate here. The experience of Sakhalin Energy has become almost 'textbook' in the literature on corporate social responsibility and Indigenous peoples. The experience of the program Sakhalin Indigenous Minorities Development Plan, implemented since 2006, has been described in detail in the academic literature [3,23–26]. Perhaps the difficulty of evaluating a company's activity is connected precisely with the breakthrough nature of its policies. Its activities are far ahead of those of other companies, which today may even create difficulties, as it has no one to compete with. Another successful company in this area is Exxon Neftegaz Limited.

The corporate social responsibility of international companies on Sakhalin is directed not only at Indigenous peoples, but at the entire local population of the region, which in itself contributes to the creation of a favorable social climate. These companies have special tripartite agreements (with regional authorities and non-governmental organizations of Indigenous peoples), on which an extensive network of coordinators works. During the field study of these companies, I got the impression that a main difference is that Sakhalin Energy has a 'Sakhalin Indigenous Minorities Development Plan', which is a clear document with prescribed organization and control mechanisms. Moreover, it is important to emphasize that the distribution of funds is carried out exclusively by representatives of the Indigenous community. The main significance of the plan is that it is aimed at the sustainable development of the Indigenous peoples of Sakhalin and contributes to the formation of social capital.

The third stage of the implementation of the development plan is underway. In 2016–2020 its objectives were development of potential (improvement of leadership qualities and technical skills; support for the pursuit of the further development of ethnic identity); socio-economic and cultural development (cultural revival, economic sustainability of enterprises engaged in traditional management); and improvement of social conditions as targeted areas of support. Emphasis is placed on long-term strategic planning with the concept of sustainable development (as a guideline); preparations for the creation of an independent fund for the development of Indigenous peoples; and

information disclosure about the environmental impact of the Sakhalin-2 project (ensuring timely, objective, and complete information about the existing and or potential impact and measures taken to prevent and or minimize any possible negative impact). Since 2006, representatives of Indigenous peoples and their organizations proposed more than 650 projects in support of traditional economic activities and implemented social development, for which the Indigenous people themselves approved the funding. The benefits of the plan include its duration and transparency.

In a comparative analysis of the policies of industrial companies with regard to Indigenous peoples in the Russian Federation, the plan stands out favorably for its focus on the development of these peoples. These are projects for the purchase of equipment and the equipping of their camps and hunting areas. Such development plan, as a policy of interaction between Indigenous peoples and an industrial company today needs not only to be encouraged, but also improved. O.A. Bazaleev, who worked for Sakhalin Energy for a long time, conducted an interesting analysis of social capital as a factor in the implementation of social and economic programs. The role of the concept of social capital is especially important for analyzing the effectiveness of development projects, as it actualizes such a resource as the interrelations between people and public structures and, "allows to avoid a one-sided perception of aboriginal communities through the prism of scarcity of resources." However, it is necessary to take into account that social capital can have an ambivalent effect: Costs such as ethnocentrism or 'clanhood' can become an obstacle to the development of Indigenous peoples [27]. The development plan covers all the Indigenous peoples of the island, regardless of where they live. The plan creates prerequisites for the increase in level and quality of life of Indigenous communities. In general, an important positive factor is the wide range of activities of international companies on Sakhalin, aimed at the entire local population of the region. But the special rapporteur emphasized that despite the apparent success of the Sakhalin agreement, there remain many problems in the relationship, including the oil producer's fulfilment of its obligations under the agreement [10]. Initially, one of the criteria for the plan's effectiveness was supposed to be, "the number of Indigenous representatives employed in the project as compared to the total workforce." Today, the company has desisted from the plan (First Sakhalin Indigenous Minorities Development Plan 2006, author archive).

*3.3. Who Enforces Company Policies and How?*

In relations with companies, it is necessary to take into account the state of psychological health of Indigenous peoples, who often experience stress from being close to oil and gas workers, etc. Hostile relationships are usually based on a lack of knowledge. Many oil workers, who have been working in the north for decades, say with an incomprehensible pride in their voices that they have never seen Natives and do not seek to learn about their life and culture. Oil and other industrial workers have formed groups which now exist as separate communities, not having and not seeking to have connections with the local population. In recent years, the policies of some companies are changing: They are starting to think more about corporate social responsibility and the adoption of special regulations for Indigenous peoples.

Company policies regarding Indigenous peoples are largely dependent on the general legal and political situation in the country. Today, the interests of Indigenous peoples and the local population are often secondary in making decisions about the priorities of territorial development; the social value of industrial development is not considered in the mixture of all the effects of resource development [28]. It is very important to have the free, prior informed consent (FPIC) of the Indigenous peoples to make decisions and the right, guaranteed by law, to say **"no"**. At the same time, field research shows that, in practice, the work of companies and the situation of Indigenous peoples in industrial areas vary.

An important indicator is the education and professional competence of staff responsible for working with Indigenous peoples. When conducting field research on Russian and international companies and when these companies create departments to work with Indigenous peoples and local people, the differing approach to recruitment is striking. In the mid-1990s, when I began my work, such a department in Kogalym (LUKOIL—Western Siberia) named a committee 'On Corrosion of

Pipes and Work with Indigenous Peoples'. In the 1990s, companies entrusted this work to engineers who had Indigenous assistants. Neither had any special education. Today, in Russian companies, people without special education continue to work in this area; they believe their activities are aimed at helping Indigenous people. In international companies, such departments pursue their activities in accordance with international principles aimed at the development of Indigenous peoples. They usually employ professionals with a humanitarian education.

If we compare the situation during the last 30 years, there is unconditional progress in relations between Indigenous peoples and industrial companies. As one indicator of this, we may consider the regulations imposed on company employees. Restrictions on hunting, fishing, gathering of wild-growing plants, keeping dogs, weapons, etc., have appeared. These rules are fairly simple to implement, and companies, using administrative resources and penalties, can easily regulate them. Such regulations usually address the accumulated, often negative experience and serve primarily to prevent conflicts. However, they contribute little to the development of partnerships.

Regulations can be the subject of wide discussion. For example, in the KhMAO, after a 2015 conference on 'Indigenous peoples. Oil. Law', ethics recommendations commenced in May 2016, for workers of industrial companies and Indigenous people when working in territories of traditional nature use. An important feature was the reciprocity of requirements. Based on such documents, companies develop policies, which they then consider when developing regulations for employees. Municipal authorities also try to take into account the specificities of companies in the Arctic in their documents on the organization of public hearings (YaNAO). Of course, all these documents, as well as laws protecting the rights of Indigenous peoples, appeared due to these efforts.

The situation is more difficult with daily work. A comparison of international and Russian companies shows that the special education of such workers, their training and retraining, and the continuous improvement of their qualifications are important, as we see, for example, in Exxon Neftegas. Another successful practice is the (ethno-sociological) monitoring, which Sakhalin Energy conducted. Russian companies, for example Lukoil-Western Siberia and Surgutneftegaz, are more likely to continue their old way, relying most often on engineers or Indigenous persons, with little thought about their qualifications.

The most important indicators of the success of the development and operation of such regulations are their legitimation, openness and provision of feedback mechanisms. In practice, we are faced with the fact that often not only the Indigenous persons and ordinary workers, but even those employees who are responsible for working with Indigenous people, do not know about these regulations. Obtaining such documents is also not without incidents: sometimes they are declared closed to the public. Only Sakhalin Energy has a clear complaint procedure, while other companies often provide general telephone numbers that Indigenous people are invited to call in case of problems. In addition, the text of the regulations is often limited to minimal restrictions and statements about respect for the culture of Indigenous peoples.

The effectiveness of company policies also depends on how its employees perceive corporate social responsibility. This is also manifested in their dissatisfaction with payments to the Indigenous people, and in their desire to not advertise some of their achievements in terms of signing and accepting documents, so that Indigenous people cannot refer to these and make new demands. A study of the opinions of managers of oil and gas companies enrolled in MBA programs collected interesting materials. These managers listed the top five areas of concern, which included policies aimed at developing the potential of companies, creating jobs, and developing sports. The interests of Indigenous and local populations as part of CSR scored substantially fewer points. The authors of this study conclude that the perception of CSR by Russian oil producers is, "a rather motley mix of Soviet heritage, memories of the turbulent 90s, today's realities and Western approaches that were relatively recently introduced into Russia" [29] (p. 28).

## 4. Discussion and Conclusions

*4.1. How do Indigenous People Perceive Corporate Compensations—As a Fair Reimbursement of Damages, a Partnership, or an Unemployment Allowance?*

Indigenous peoples have been entitled to compensation for damages that industrial companies caused since 1999, according to an article incorporated in the Federal Law 'On Guarantees of Rights of Numerically Small Indigenous Peoples of the Russian Federation'. Why is it so difficult to realize this right in practice? There are a number of factors limiting this right of Indigenous populations. Firstly, it does not correspond to company responsibilities, and is not covered in the law on sub-surface resources. Secondly, this right should be built upon the free prior and informed consent (FPIC) of Indigenous peoples regarding decisions relating to their interests, as declared in international documents. Companies need to realize that compensation is one condition for obtaining Indigenous people's consent. Field research shows that for the just realization of the right to FPIC representation of Indigenous peoples is required at consultations and public hearings. Moreover, persons having a dissenting opinion should be entitled to court appeal, supplementary expertise, etc.

Availability of financial means is a very important condition to allow Indigenous populations to make decisions—they should be entitled both to exercise control over resource use, and have the right to priority access to resources, firstly to fishing and hunting lands and reindeer pastures, as the basis of their livelihood. Indigenous economies consist of not only state support and compensation payments from industries, but also their guaranteed right to free use of natural resources. Only then, Indigenous people will feel free to resolve issues concerning the use of the territories which they inhabit.

Today, two lines of compensation for damages experienced by Indigenous populations exist in the Russian Federation. Naturally, compensations cannot be a panacea for all the inadequacies of industrial development, however they may alleviate and minimize the consequences of such damages. First are financial payments, based on agreements, or on the basis of various methods of calculation. The result is the receipt of money, or sometimes other things of material value. This practice is best exemplified by KhMAO (among regions under my research). At the beginning of the 1990s, it was pioneering, which was a breakthrough in Russia. Unfortunately, as it often happens, its positive value has diminished over time; agreements became increasingly bureaucratized towards more formal identification of conditions instead of improvement and enhanced opportunities. These agreements very quickly ceased to restrict the activities of the companies. The Indigenous people were in a difficult situation. They are financially dependent on these compensations, as their ability to practice their own economic activities diminished. In the areas designated for traditional nature use there are more and more new industrial facilities. Sometimes the Indigenous people have to become shift-workers, and sometimes this is in close proximity to their own ancestral land. Such neighbors certainly reduce the possibilities for traditional nature use. This system often does not suit either party. Much is contextual, depending on personal relationships, not on law and justice [14] (p. 72). There is a danger that the developed method of calculating losses will change little. Experts (economists) are primarily concerned with the accuracy of formulas. The Committee on Issues of Nationalities of The State Duma does not support the proposals to discuss the procedure for using the methodology in discussions.

What happens in the areas of industrial development as a result of financial injections from large businesses? How do Indigenous peoples' lives change? How satisfied are they with the changes? Indigenous experts suggest their own visions of compensation, which are not only material or financial. Here is a notable quote from an Aboriginal teacher and UN 'Unsung Hero of the 20th Century' on Presentation to a San-!Khoba project workshop (Kalk Bay, South Africa, June 2006), cited in the 2009 book *Indigenous Peoples, Consent and Benefit Sharing: Lessons from the San-Hoodia Case*: "My biggest advice would be, please, … do … not … just focus on the economic gains, because for Indigenous people the most important thing is the relationship" [9] (p. 334). Many Indigenous residents of the Russian North also speak in favor of safeguarding the environment and preserving accessible renewable resources (primarily land) as well as the opportunity to support their families.

Seminars revealed interesting results in Bulunsky Ulus of the Sakha Republic (Yakutia) in 2018, especially during the training seminar: 'What do Indigenous peoples expect from ethnological expertise and how can we achieve this?' The seminar made use of the method of 'wheel of balance'. While this is useful for personal improvement, in our case, the psychologist-coach T.F. Martynova

applied it to work with groups. During a role-playing game, participants were divided into 'Indigenous people', 'industrialists', 'local population and NGOs', and 'government bodies and local governments', and we tried to ensure that participants of the seminar choose roles contrary to their life experiences. Compared with the usual 'wheel', which includes important segments for each person (e.g., family, health, finances), instructors described sections that characterize traditional culture, language, and ecology as the most relevant for Indigenous peoples. Participants had to present their satisfaction in points from 1 to 10. This technique has proven successful; people started thinking about how to build relations with companies and about what they expect from ethnological expertise. Participants made very important conclusions that employment is more important than compensation. It should be noted that participant point of view took shape during the course of the seminar; at the beginning, everybody had said that the company should build and do everything for them and provide money. The second important point of the seminar is related to acquiring the experience of communicating on an equal footing in partnership with companies and authorities. This is especially important in modern conditions for regions where industrialists and Indigenous peoples become neighbors, who not only work, but also live side by side.

The competent social and environmental policies of companies aimed at the sustainable development of Indigenous peoples determine the other line is determined. In Russia today, it is represented by the Sakhalin Indigenous Minorities Development Plan (not to say this cannot be improved). It also has a financial component, but there is also a mutual understanding that "a gift of a fishing rod is more important than a gift of a fish". It is also important to note the emerging new opportunities for the development of Indigenous business. The plan allocates grants and microcredits for the purchase of equipment, transportation, and arrangement of processing points for products. Recipients can spend funds on the purchase of autonomous power plants, information and communication technologies, and electrical appliances useful for commercial activities.

### 4.2. Possibilities for Legal Regulation of Corporate Social Responsibility

The experience of studying the legal regulation of the interaction of Indigenous peoples and industrial companies shows its inadequacy for establishing equitable partnerships and co-management. Today, a new platform is forming to solve this problem—a draft law on public non-financial reporting and a plan for its implementation. In May 2017, the Government approved the "concept of public non-financial reporting", prepared a corresponding draft law, and created an interdepartmental group to develop a list of key non-financial reporting indicators [30]. The Expert Council of the Committee on Issues of Nationalities of The State Duma began negotiations on the possibility of taking into account the interests of numerically small Indigenous peoples in this process.

Until recently, local communities perceived the reports on non-financial reporting and corporate social responsibility as showing voluntary activities of companies. However, research shows the shortcomings of these policies: They are opaque, they focus on a narrow group, and far from all companies operating in the North and the Arctic generate sustainability reports. Citizens' passivity is also an obstacle [31]. Among Indigenous populations, this may be attributed to their low level of education, especially in legal matters, their low living standards, lack of material resources, and the resulting high degree of dependence on financial support from industrial companies, especially in KhMAO.

In their social policies, international standards and Russian standards have adapted to these guide companies. It is not by chance that new opportunities for including the interests of Indigenous peoples on the agenda appeared precisely in the summer of 2018. On 1 July, the GRI 411—Indigenous Peoples (Global Reporting Initiative) entered into force [32]. The development of formal criteria for non-financial reporting in Russia today is of interest to all. Large companies are interested in increasing their investment attractiveness, Indigenous people want their voice to be heard. Companies are interested in receiving feedback from the regions of their activities, while Indigenous people want to know what awaits them during and after the implementation of industrial projects.

Yet, with respect to Indigenous peoples, company policies lack transparency and partnership. Some of them invest heavily, but do not see returns in society.

According to the GRI 411 standard, non-financial reporting requires companies to identify cases of recorded violations of the rights of Indigenous peoples. Unfortunately, the clarifications of this document are not taken into account in Russia. In international practice, it is included in the system of protection of the rights of Indigenous peoples. Two non-financial reporting indicators are included in the Russian version—the number of formal cases concerning the numerically small Indigenous people's violation of rights and the number of internal corporate documents in the company on policies regarding these people. The Expert Council of the Committee on Issues of Nationalities of The State Duma included a more detailed description of non-financial indicators and offered to decipher the conflict resolution mechanism in the light of the complaint procedure. Experts also proposed to introduce additions to the index on 'characterization and training of personnel' for the employment of Indigenous people, including training and retraining programs.

The interaction between Indigenous peoples of the North and industrial companies is certainly a complex and intractable problem to solve. The sheer proximity of various actors adds to its severity. Climate change, reduction of renewable resources, the need to improve the level and quality of life of the population in areas of industrial development dictate a clearer position for the state and civil society. Corporate policies of companies are becoming a promising field for dialogue. Experts proposed various means of legal regulation of these relations including international law, Russian legislation, and standards of business communities. It is obvious that the Indigenous peoples themselves and their organizations play a significant role. Their participation can suggest best practices and the mechanism for their implementation in specific regions for specific companies. Indigenous involvement can also offer possible mechanisms for the prevention and resolution of conflicts.

Benefit sharing of oil and gas production is a long process. In order to be successful, it requires legal regulation, everyday work of the companies, supervision and involvement of Indigenous peoples, as well as stakeholders' observance of mutual agreements. Applied anthropological research has demonstrated that companies are to combine equitable compensations covering the actual damages done to Indigenous peoples with socially responsible policies and high international business and human rights standards.

**Funding:** This research was funded by Russian Foundation for Basic Research (RFBR, grant No. 18-05-60040) written in the framework of the project 'New technologies and social institutions among Indigenous peoples of the Russian Arctic: Opportunities and risks'.

**Acknowledgments:** I am very grateful to three anonymous reviewers for their positive and constructive comments on the first submission of this paper. I would also like to express my thanks to all the many respondents I engaged with in the field. I would also like to thank all my colleagues with whom I worked on projects on interaction between Indigenous peoples and industrial companies in Norway, Canada, and Russia. I would also like to thank Maria Tysiachniouk, with whom I have exchanged many ideas around the topic of benefit sharing and who encouraged me to submit this paper to this collection.

**Conflicts of Interest:** The author declares no conflict of interest.

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
