# Peer review of "Pipeline Neighbors: How Can We Avoid Conflicts?"

_resources, doi:10.3390/resources9020013_

Round 1

Reviewer 1 Report

The paper is very interesting and I am not one to comment on what is going on in Russia as I have not had much experience there. Having worked largely in the circumpolar region, northern Canada, for Indigenous organizations, I would have perhaps oriented the arguments a bit differently, but this has more to do with style than content.

The paragraph about Canada on page 2, is accurate but would be more accurate to point out that Indigenous peoples have had to advance their rights and interests through the Canadian courts, applying the Constitution of 1982 and Charter of Rights and Freedoms. Government has not been benevolent, companies have not been proactive partners.

On page 7 one statement is very important about how companies and Indigenous peoples see justice differently. I would add a long list of how diverse even each of the constituencies are, that they also have very different conceptions of what constitutes land, resources, connections to the spiritual and cultures, a good life, etc.

UNDRIP is not mentioned at all, is there a reason for that?

on page 4 the interview data should be unpacked, how many interviews, what were the genders and ages of those who are key informants? Gender plays a major role in perceptions and should be discussed.

on page 5 there is the assumption that being a neighbor is a good thing and mutually advantageous, but respect doesn't come from being forced to watch the land transformed with mining, hydro, oil and gas development. So I wonder what this means and that is pretty much the crux of my criticism of the paper, there seems to be some unspoken assumption about partnerships that are forced on Indigenous peoples to develop. I am sure the author appreciates this idea and the challenges it makes to the analysis.

I am going to accept with changes, and leave it up to the author to decide what to do with my review. As I stated to begin with, I am not in Russia and I don't know the political and economic climate of scholarship and would accept that some of what I have raised is easily raised in Canada and elsewhere. This is not to say that Canada isn't problematic when it comes to impacts from development on Indigenous lands and peoples, we are home to cultural genocide that is on going.

Author Response

Thank you very much for your review. It is very important for improving this article. I took into account all your recommendations.

Reviewer 2 Report

This is a well-written, interesting, and useful overview and analysis of the relationship between industrial actors and indigenous people, primarily focused on the Russian Arctic, but informed by international laws and other Arctic cases.  Particularly valuable elements of the article include the combination of legal analysis of policies at many levels (International/national/regional) and extensive ethnographic field work as well as the comparison across regional cases in Russia. The article is best when the author backs up broad generalizations with specific examples from these cases. The author has done a very fine job of identifying both challenges and opportunities that relate to avoiding conflict between indigenous communities and oil and gas companies and tensions between compensation and sustainable development within the rubric of benefit sharing. 

Overall, I found the article to be very well done and I recommend publication.

I have just a few minor suggestions:

--The author talks about legal pluralism in the abstract and theory section of the article, but the concept is not really mobilized in the discussion and conclusion.  Legal pluralism implies complexity and overlap among policies that can both empower actors to choose which laws to comply with and to generate confusion among actors.  I see several opportunities to highlight this point in the discussion, linking the cases more closely to the theory for authors who do not work in Russia.

--Also in the abstract, the author mentions “the benchmark for the comparison’ as international and Russian standards.  I didn’t really see this type of benchmarking in the paper.  It is implied by references to FPIC and other global indigenous rights, but not made specific.  I would beef this up or cut from the abstract.  Retaining it, but developing it would help us understand what the author means by “successful” in the conclusion (now used on p. 13 in reference to the seminar in Yakutia).

I also found a few minor errors in the text:

--Sentence fragment on page 8: “Negative impact on the situation of indigenous peoples and narrowing the 344 scope of legal regulation of companies in the region.”

--Page 9 – needs references: “The 367 experience of Sakhalin Energy has become almost ‘textbook’ in the literature on corporate social 368 responsibility and indigenous peoples. The experience of the program “Sakhalin Indigenous 369 Minorities Development Plan” implemented since 2006, has been described in detail in the academic literature.”

-- Page 10 – repeated text: “In the mid-1990s, when I began my 442 work, such a department in Kogalym (LUKOIL – Western Siberia) was called “On Corrosion of 443 Pipes and Work with Indigenous Peoples”. In mid-1990s, at the start of my research, such 444 department in Kogalym (LUKOIL–West Siberia) was titled ‘Pipe corrosion and Indigenous people’s 445 relations,’ In the 1990s, companies entrusted this work to engineers who had indigenous assistants.”

--Page 12 – text almost identical so need to clarify different meaning: “They should 510 have the right not only to control the use of resources, but the right of priority access to these, firstly 511 to fishing and hunting lands and to reindeer pasture, as the basis of their livelihoods. It is important 512 that their domestic economies are constructed not only on state support and compensation 513 payments from industries. 514 Availability of financial means is a very important condition to allow indigenous populations make 515 decisions – they should be entitled both to exercise control over resource use, and have the right to 516 priority access to resources, firstly to fishing and hunting grounds and reindeer pastures, as the basis 517 of their livelihood.”

A well done article!

Author Response

Thank you for your review. It is very important for improving this article. I took into account all your recommendations.

Reviewer 3 Report

 apparent typing error in line 540

the article is well written, appears to be scientifically solid and I am sure that it will be of interest for your regards. The text might benefit from expanding the introduction a bit to make the subject matter more accessible to regular readers of "Resources" who might not be familiar with that specific situation.

Author Response

Thank you very much for evaluating my article. I included additional literature in to introduction. Your comment concerns the name of the Russian agency.

Committee on Issues of Nationals of the State Duma is the official name of the Committee of the lower chamber of parlament.